# Expression of miR-135b in Psoriatic Skin and Its Association with Disease Improvement

**DOI:** 10.3390/cells9071603

**Published:** 2020-07-02

**Authors:** Pablo Chicharro, Pedro Rodríguez-Jiménez, Mar Llamas-Velasco, Nuria Montes, Ancor Sanz-García, Danay Cibrian, Alicia Vara, Manuel J Gómez, María Jiménez-Fernández, Pedro Martínez-Fleta, Inés Sánchez-García, Marta Lozano-Prieto, Juan C Triviño, Rebeca Miñambres, Francisco Sánchez-Madrid, Hortensia de la Fuente, Esteban Dauden

**Affiliations:** 1Dermatology Department, Instituto de Investigación Sanitaria Hospital Universitario de la Princesa (IISP), 28006 Madrid, Spain; somniem@gmail.com (P.C.); pedro.rodriguez.jimenez90@gmail.com (P.R.-J.); mar.llamasvelasco@gmail.com (M.L.-V.); estebandauden@gmail.com (E.D.); 2Rheumatology Department, Instituto de Investigación Sanitaria Hospital Universitario de la Princesa (IISP), 28006 Madrid, Spain; nuria.montes.casado@gmail.com; 3Fisiología Vegetal, Departamento Ciencias Farmacéuticas y de la Salud, Facultad de Farmacia, Universidad San Pablo-CEU, 28003 Madrid, Spain; 4Data Analysis Unit, Instituto de Investigación Sanitaria Hospital Universitario de la Princesa (IISP), 28006 Madrid, Spain; ancor.sanz@gmail.com; 5Immunology Department, Instituto de Investigación Sanitaria Hospital Universitario de la Princesa (IISP), 28006 Madrid, Spain; danay.cibrian@gmail.com (D.C.); alicia.vara@salud.madrid.org (A.V.); jmzfdzmaria@gmail.com (M.J.-F.); martinezq93@gmail.com (P.M.-F.); ines.sanchez.garcia9589@gmail.com (I.S.-G.); mlozanoprieto54@gmail.com (M.L.-P.); fsmadrid@salud.madrid.org (F.S.-M.); 6CIBER de Enfermedades Cardiovasculares, Instituto de Salud Carlos III, 28009 Madrid, Spain; 7Bioinformatics Unit, Centro Nacional de Investigaciones Cardiovasculares (CNIC), 28029 Madrid, Spain; manueljose.gomez@cnic.es; 8Sistemas Genómicos, 46980 Valencia, Spain; jc.trivino@sistemasgenomicos.com (J.C.T.); rebeca.minambres@sistemasgenomicos.com (R.M.)

**Keywords:** psoriasis, miR-135b, treatment response, miRNAs

## Abstract

miRNAs have been associated with psoriasis since just over a decade. However, we are far from a complete understanding of their role during the development of this disease. Our objective was to characterize the cutaneous expression of miRNAs not previously described in psoriasis, the changes induced following the treatment with biologicals and their association with disease improvement. Next generation sequencing was performed from five skin samples from psoriasis patients (lesional and non-lesional skin) and five controls, and from this cohort, 12 microRNAs were selected to be analyzed in skin samples from 44 patients with plaque psoriasis. In 15 patients, an additional sample was obtained after three months of biological treatment. MiR-9-5p, miR-133a-3p and miR-375 were downregulated in the lesional skin of psoriasis patients. After treatment, expression of miR-133a-3p, miR-375, miR-378a and miR-135b in residual lesions returned towards the levels observed in non-lesional skin. The decrease in miR-135b levels after treatment with biologics was associated with both the improvement of patients evaluated through Psoriasis Area and Severity Index score and the decrease in local inflammatory response. Moreover, basal expression of miR-135b along with age was associated with the improvement of psoriasis, suggesting its possible usefulness as a prognostic biomarker.

## 1. Introduction

Psoriasis is a very complex disease that results from the interplay between the immune system, epidermal cells, genetic and environmental factors. The interaction between immune cells and keratinocytes leads to a self-amplifying response that results in the presence of psoriatic plaques characterized by keratinocyte proliferation and recruitment of different leukocyte subsets into the skin [1].

MicroRNAs (miRNAs) are highly conserved small non-coding RNAs with a key role in the regulation of gene expression [2]. Our knowledge about many biological functions of miRNAs comes from studies of overexpression or blockade of individual miRNAs. However, studies from knockout animals have shown that the absence of a single miRNA results in modest or no clear alteration of phenotype, supporting the notion that miRNAs do not work in isolation, but as part of regulatory networks of gene expression [3,4,5]. An additional feature of miRNAs that adds complexity to their function is that one single miRNA is able to regulate the expression of many genes, each of them being a possible target of many miRNAs [6].

During inflammation, the miRNA expression is modified and several proteins induced along the process, including p53 and IFN-gamma, are able to regulate miRNA processing [7,8]. It is also known that the expression of miRNAs is sex-biased [9,10], and also influenced by age [11] or lifestyle factors [12]. All these data empower the overall study of miRNAs in human inflammatory pathologies.

Psoriasis, as other chronic autoimmune diseases, shows a specific profile of miRNA expression. Numerous miRNAs are dysregulated in the skin of psoriasis patients, with miR-21, miR-31, miR-146a, miR-155 and miR-125b among the most frequently described [13,14]. Interestingly, the polymorphism rs2910165 in miR-146a has been described as a risk factor for psoriasis [15,16]. However, we are far from having a complete catalogue of dysregulated miRNAs in this pathology.

The main objective of this work is to characterize in patients with psoriasis the cutaneous expression of miRNAs not previously associated with this pathology. Furthermore, we assess the influence of demographic and clinical factors on the expression of newly identified miRNAs as well as that of miR-31-5p, miR-378a, miR-135b, miR-142-3p and miR-146a-5p (previously associated with psoriasis). Finally, we evaluate the expression of miRNAs after the treatment with biological agents and their association with disease severity and the improvement of the Psoriasis Area and Severity Index (PASI) after treatment.

## 2. Materials and Methods

For a more complete detail of Material and Methods see Appendix A.

### 2.1. Study Subjects and Sample Collection

The study was approved by the independent ethics committee of the Hospital Universitario de la Princesa and performed according to the Declaration of Helsinki principles. Small RNA NGS assays (SOLiDv4) were performed with both lesional (L) and non-lesional (NL) skin samples from 5 patients with moderate-to-severe plaque psoriasis as well as with healthy (H) skin from 5 control subjects. Data from NGS were then validated in 44 additional psoriasis patients. All patients had a Psoriasis Area and Severity Index (PASI) ≥ 10. Before obtaining skin samples, the following washout periods were established: 28 days for conventional systemic treatment; and 84 days for biological agents. Skin punch biopsies (4 mm) were obtained from psoriasis plaque and non-lesional skin from at least 5 cm from the closest plaque; samples were taken preferentially from non-sun-exposed areas. In 15 out of 44 patients, an additional skin sample was obtained after 3 months of treatment (anti-IL-12/23 *n* = 8, anti-TNFa *n* = 4, anti-IL-17 *n* = 3). In these cases, the skin biopsy was taken from an area with a previous plaque that had resolved after treatment.

### 2.2. Small RNA NGS

NGS was performed using the platform SOLiD v4 at Sistemas Genómicos (Valencia, Spain). Quality and quantity of RNA were evaluated using Agilent 2100 Bioanalyzer and Qubit 2.0 fluorometer. Libraries were prepared following the protocols of Life Technologies for SOLiD v4 sequencing. Quality of libraries was evaluated using Qubit and Agilent 2100 Bioanalyzer. Differential expression analysis with a ±2-fold change was performed to compare the expression of known miRNAs that were identified by BedTools and miRDeep2.

### 2.3. Tissue miRNA Isolation, Reverse Transcription and RT-PCR

Expression of miRNAs was analyzed using miRNA LNA PCR primer sets (Exiqon, Aarhus, Denmark). Genorm algorithm (part of Biogazelle qbase+ software) was used to identify the most stable reference genes. Data were normalized using the geometric mean expression of 5S ribosomal RNA and RNU1A1 identified as most stable. Thus, data are expressed as the relative levels of indicated miRNAs with respect to the geomean of 5S and RNU1A1. Differential expression of miRNAs in non-lesional and lesional skin samples was tested using Wilcoxon signed rank test. The expression of miRNAs was fitted to a normal distribution according to Shapiro–Wilks and Akaike’s information criteria (AIC) (R package: riskDistributions).

### 2.4. miRNA Target Profiling

Identification of miRNA targets was performed after reanalyzing public gene expression data from a study in which the transcriptomic profile of lesional and non-lesional skin samples from 14 psoriasis patients was compared by RNASeq GEO (GSE67785) [17]. Interactions between miRNAs and differentially expressed genes were identified with IPA tool “microRNA Target Filter” (Ingenuity Pathway Analysis, Qiagen, Hilden, Germany). First, miRNA identifiers were used as query to identify mRNA partners supported by experimental evidence or predicted with high confidence. Then, interacting pairs were filtered to keep only those with anti-correlated expression in lesional vs. non-lesional skin samples. The resulting collection of genes was subjected to functional core analysis with IPA, to identify associations to canonical pathways, upstream regulators and diseases or biological functions. As an additional approach, miRNAs targets were identified using the miRTarBase software selecting only those classified as “Functional”, then the list of genes was subjected to the PANTHER Classification System (http://pantherdb.org) to identify association with biological processes. As a reference list, DE expressed genes in lesional psoriatic skin vs. non-lesional skin from GEO (GSE67785) were used.

### 2.5. Statistics

Effect of demographic and clinical parameters on the expression of miRNAs was analyzed by a general linear model (GLM). Differences between groups were compared using paired t-test or Wilcoxon signed rank test as appropriate. A transformation was applied to miRNA expression to achieve a normal distribution in order to apply parametric tests. Multiple regression analysis was used to determine the association of miRNAs values with disease severity evaluated by PASI. Ordered multivariable logistic regression was performed to analyze the relation between miRNA expression in lesion or non-lesion and improvement evaluated by PASI score. Bonferroni correction was applied for multiple comparisons. The *p*-values were two sided and statistical significance was considered as *p* < 0.05. Statistical analyses were conducted using R version 3.6.1 (https://www.R-project.org/).

## 3. Results

### 3.1. miRNA Profiling of Psoriatic Skin by NGS

To identify candidate miRNAs not previously described as differentially expressed in psoriatic patients, NGS was performed from five psoriasis patients (lesional and non-lesional skin) and five healthy subjects. Forty-nine miRNAs were differentially expressed in at least one comparison: healthy (H) skin vs. non-lesional (NL) skin, H vs. lesional (L) skin or L vs. NL skin with a Bonferroni adjusted *p*-value < 0.05. The dendograms of a heatmap show a greater similarity between the samples of healthy subjects and non-lesional skin of psoriasis (Figure 1a). Thirteen out 49 miRNAs were differentially expressed (± 2-fold) between NL and H skin, while a total of 23 miRNAs were differentially expressed in L vs. H and 22 in L vs. NL (Appendix A and Figure 1b).

From the NGS data, we selected 12 miRNAS to be studied in skin samples from the larger cohort of 44 patients, using RT-PCR. From these 12 miRNAs, we selected 8 MicroRNAs as representatives of each comparison group that had not been previously validated in psoriasis: miR-9-5p, miR-375 and miR-33b DE in L vs. NL skin; miR-3145, miR-3687 and miR-934 DE in L vs. H skin and finally miR-133a-3p and miR-614 DE in NL vs. H skin (Appendix A). Four miRNAs were included as controls because of their relevance in psoriatic skin inflammation (miR-31-5p, miR-135b-5p, miR-378a, miR-142-3p) [13,18]. Although no differences were detected in the expression of miR-146a-5p, this miRNA was included in the validation phase as an additional control because of its involvement in psoriasis [19,20].

### 3.2. Effect of Demographic and Clinical Variables on miRNA Expression

The influence of demographic and clinical variables on miRNA expression was analyzed in the 44-patient validation cohort, for the subset of 12 species selected on the basis of NGS data analysis. Demographic and clinical characteristics are summarized in Appendix A. A higher expression of miR-133a expression was detected in smokers compared with non-smoker patients (*p*-value = 0.05, β-coeff = 0.42). Regarding gender, only the expression of miR-133a showed differences between female and male, being the expression of this miRNA higher in the latter group (*p*-value = 0.03, β-coeff = 0.45). Our data showed that age was a demographic variable that affected miRNA expression. Hence, higher levels of miR-142 (β-coeff = 0.412, *p*-value = 0.043) and miR-378a were detected in those patients over 45 years (β-coeff = 0.470, *p*-value = 0.034), while miR-135b expression was diminished in older patients (β-coeff = −0.528, *p* = 0.043).

On the other hand, we did not detect any differences in miRNA expression associated with the presence of psoriatic arthritis (PsA). Dyslipidemia was associated with higher levels of miR-375 (β-coeff = 1.224, *p*-value = 0.001), and lower levels of miR-146a (β-coeff = −0.969, *p*-value = 0.038) and miR-3145 (β-coeff = −1.194, *p*-value = 0.027). Differences in miR-146a (β-coeff = 1.020, *p* = 0.010) and miR-375 (β-coeff = −1.110, *p*-value = 0.001) were also observed in DM. Finally, lower levels of miR-135b (β-coeff = −0.703, *p* < 10^−3^) were detected in those patients with arterial hypertension (Figure 2).

MiRNA-615 and miRNA-934 were excluded because their expression was undetectable (data not shown).

### 3.3. Differential Expression of miR-9-5p, miR-375, miR-133a and Their Modulation Following Treatment

Our data showed that psoriatic lesion skin expressed lower levels of miR-9-5p, miR-133a-3p and miR-375 compared with non-lesional skin (*p*-value < 0.050, Figure 3a). On the contrary, higher levels of miR-135b-5p, miR-378a, miR-142-3p, miR-146a-5p and miR-31 were detected in lesional skin (Figure 3b). A heat map of hierarchical clustering of skin samples based on the expression of validated miRNAs is shown in Figure 3c. No significant differences were observed for miR-3145, miR-3687, and miR-33b (data not shown).

Next, we sought to find out whether the expression of these miRNAs changes after treatment. To check it, we compared miRNA expression in lesional skin vs. the residual lesion in 15 patients treated with biologics (Appendix A). After three months of treatment, when local inflammation had significantly decreased as demonstrated by the expression of *IL-12b* and *S100A9* in residual lesions (Appendix A), the expression of miR-375 and miR-133a-3p increased while the expression of miR-378a and miR-135b diminished, thereby returning to those levels detected in non-lesional skin (*p*-value < 0.050, Figure 4a). On the contrary, no differences in the expression of miR-31, miR-146a, miR-142 and miR-9-5p were detected between lesional skin and residual lesions (Figure 4b). Interestingly, the decrease in mir-135b expression was significantly associated with patients’ rates of improvement according to PASI as indicated by the correlation of the disease improvement after three months of treatment with miR-135b expression in residual lesions (Spearman r = −0.6 and *p*-value = 0.02) (Figure 4c). Moreover, miR-135b levels in residual lesions were clearly associated with the expression of *S100A9*, an indicator of inflammation (Spearman r = 0.86, *p* value = 0.0006) (Figure 4d).

### 3.4. Relationship between miRNA Expression and Psoriasis Activity

To analyze the relation between miRNA expression in lesional or non-lesional skin and psoriasis severity, we identified the clinical and demographic variables associated with PASI. Statistical analysis revealed that DM and basal levels of creatinine were associated with disease activity (*p*-value = 0.073, β-coeff = −7.193 and *p*-value = 0.070, β-coeff = 16.280, respectively, data not shown). Then, the association of miRNA expression and PASI was analyzed adjusting for DM and creatinine levels. Among the 12 miRNAs analyzed, only the expression of miRNA-9-5p in lesional skin was associated with PASI values (*p*-value = 0.005, β-coeff = 7.35) (Table 1).

### 3.5. Basal Expression of miR-146a and miR-135b Is Associated with Disease Improvement

To analyze the association between the levels of miRNAs expressed before the treatment and the improvement of patients, we considered only those patients treated with biological drugs (*n* = 33). Fifteen patients were treated with anti-IL-12/IL23, 8 with anti-TNFa and 11 with anti-IL-17. After three months of treatment, 7 patients (21.2%) achieved PASI 100, 18 patients (54.5%) PASI 90, 4 patients (12.1%) PASI 75, 3 patients (9%) PASI 50 and only 1 patient (3%) improved 25% from baseline. First, we analyzed if the psoriasis risk factors (age, hypertension, diabetes mellitus, smoking and obesity) or the type of biological therapy used were or not associated with improvement in our patients. A significant association was detected for age (*p*-value = 0.04) and a marginal association was identified for the type of biological therapy (*p*-value = 0.13) (Appendix A). Both variables were included in a multiparametric model. Statistical analysis showed that expression of miRNA-146a in non-lesional skin (OR = 2.33, *p*-value = 0.015), miRNA-135b (OR = 6.06, *p*-value = 0.009) in lesional skin and age (OR = 0.95, *p*-value = 0.078) were related to improvement (Table 2).

### 3.6. Association of Interacting miRNA-Targets with Psoriasis

To gain insights into the biological processes in which target genes of dysregulated miRNAs could be involved, an enrichment analysis was performed using ingenuity pathway analysis (IPA). Publicly available gene expression data comparing the expression profile of L and NL skin samples from psoriasis patients were downloaded from GEO (accession number GSE67785). Interacting miRNA targets were filtered to keep only those with anti-correlated expression (i.e., high expression of miRNA in lesional skin and a lower expression of its target). Psoriasis was detected among the significantly associated diseases, with an enrichment *p*-value of 0.004 (0.069 after adjustment with the Benjamini–Hochberg method). IPA analysis showed 18 target genes associated with psoriasis (Appendix A). Since the literature review did not show direct links of most of these genes with the pathogenesis of psoriasis, we used an additional approach to identify the association of miRNA targets with this disease. Functional miRNA targets were identified using the miRTarBase software. Genes were filtered to keep only those with anti-correlated expression and subjected to the PANTHER Classification System to determine the over-representation of biological processes annotated in Gene Ontology. With this approach, we found that gene targets were enriched in immune response and angiogenesis processes (Figure 5).

## 4. Discussion

MiRNA expression profiles in different autoimmune diseases have suggested the potential of miRNA signatures, mainly circulating ones, to predict outcome or response to therapy, thus highlighting the role of these molecules as clinical biomarkers. In this sense, it has been recently described an EV-associated miRNAs signature increased in serum of psoriasis patients that returns to normal levels after successful treatment with anti-TNF-alpha [21]. Our knowledge about the expression of miRNAs in the skin of psoriatic patients after treatment is limited. Here, we identified the association of miR-135b downregulation in psoriatic skin to the improvement of PASI, while the expression of other well-known miRNAs upregulated in this condition, are not modified upon the treatment in spite of the improvement of patients.

Keratinocyte miR-31 is one of the most studied miRNAs in psoriasis; lesional skin of psoriatic skin express high levels of this miRNA [22]. In the past few years, it was described that activation of NF-κB signalling in keratinocytes induces miR-31 transcription, which regulates keratinocyte cell cycle [23]. Thus, we could expect that its levels were decreased after three months of treatment with biological agents. However, we did not detect significant differences in miR-31 levels between lesional skin and residual lesions. The expression of miR-142-3p and miR-146a did not decrease either after treatment, although it has been described that the circulating levels of these miRNA are downregulated in etanercept and adalimumab responders patients, respectively [24,25]. However, the persistence of miR-146a levels in psoriasis patients after biological treatment has been also been reported [21]. The observation that the levels of miR-31, miR-142 and miR-146a in psoriatic skin do not change after treatment with biologics suggests their association with processes that prevail upon biological treatment, as could be the case for miR-146a-5p, highly expressed by Treg cells, also fundamental for their function [26]. Nevertheless, it cannot be ruled out that the levels of these miRNAs decrease after a longer period of treatment.

Although the study of individual miRNAs is very useful, the biological function of these molecules cannot be understood as the repression of one or few targets by a single microRNA. Enrichment analysis using PANTHER System Analysis identified several targets for the dysregulated miRNAs that were significantly associated with immune response and angiogenesis processes. Several of these genes were targets for miR-9-5p, miR-133a and miR-375, supporting the notion of their involvement in psoriasis and the importance to draw a complete picture of miRNA expression in this disease. It has been very recently described that miR-9-5p protects against inflammation through the inhibition of NF-κB [27,28]. Mir-9-5p also regulates the expression of *ATG5* (Autophagy related 5) [29] and its overexpression in psoriatic skin has been recently reported [30]. Moreover, the keratinocyte-specific ablation of *ATG5* ameliorated imiquimod-induced skin lesions in mice accompanied by a reduction in the number of IL-17A producing cells [30], highlighting the role of this target in psoriasis. On the other hand, downregulation of miR-133a has been detected in asthma and its upregulation is able to ameliorate airway remodeling through PI3K/AKT/mTOR signaling pathways by targeting *IGFR1* (insulin-like growth factor I receptor [31]. *IGFR1* plays an important role in the differentiation and apoptosis of many types of cells including keratinocytes and it is overexpressed in psoriatic lesions [32]. Administration of *IGFR1* anti-sense oligonucleotides into human psoriatic lesions grafted in nude mice caused a significant improvement of hyperplastic epidermis [33]. *IGFR1* is also a validated target for miR-375 [34], which increases the possibilities that its overexpression in psoriatic skin may be associated with the lower levels of these miRNAs. The JAK2/STAT3 signaling pathway is involved in both inflammatory and anti-inflammatory signaling pathways [35] and there is solid evidence regarding the functional regulation of this pathway by miR-375 [36]. In patients with inflammatory cardiomyopathy, levels of miR133a correlate with improved cardiac function and clinical outcome [37], and the inhibition of miR-375 attenuates the inflammatory response after myocardial infarction, supporting a protective role of these miRNAs in inflammatory processes.

To identify differentially expressed miRNAs not previously validated in psoriasis, we performed NGS assays in five skin samples from psoriasis (lesional and non-lesional) and five skin samples from healthy donors. Although with this limited number of samples we were able to detect the differential expression of undescribed miRNAs in psoriasis, a larger “*n*” would have yielded differences in a greater number of miRNAs between groups and is quite possibly the reason for not detecting differences in miR-146a reported in other studies.

The only miRNA we identified, whose modulation after treatment was associated with the improvement of PASI was miR-135b. Although upregulation of this miRNA in psoriatic skin was described several years ago [14], its association with response to treatment has not been previously reported. Although the study of the functional role of this miRNA in psoriasis is beyond the scope of the present work, our results and the knowledge we have gained about the function of miR-135b indicate that this miRNA could be involved in the inflammatory process associated with this pathology. This miRNA has been mainly related in cancer and it has been associated with several functions that suggest its importance in psoriasis. For example, miR-135b promotes angiogenesis through the regulation of two targets, *FOXO1* and *HIF1a* [38,39]. FOXO1 is a transcription factor that regulates metabolic homeostasis in response to oxidative stress and has a key role in the function and development of Treg cells [40,41]. Interestingly, Treg cells from psoriasis are defective in the Akt-FOXO1 signaling pathway [42]. Dysfunctional or reduced Treg cells have been described in peripheral blood and in psoriatic lesional skin in patients [43,44,45]. The relevance of local Treg cells in skin inflammation has been recently reported, where using the imiquimod-induced murine model of psoriasis demonstrated that Treg cells limit the exacerbation of local skin inflammation and initiate also disease remission [46]. It is known that the frequency of Treg cells increases in the skin of psoriasis patients after topical treatment with steroids [47], and that anti-TNF-a drugs induce the upregulation of circulating Tregs that correlates with reduction in the disease severity score [48]. A similar finding has been recently described in a psoriasis murine model where the administration of anti- IL-17A or IL-23p19 induced a significant increase in the number or Foxp3^+^ IL-10^+^ Treg cells [49]. In addition, it is known that the blockade of miR-135b attenuates IL-17 production in anaplastic cell lymphoma [50]. Additional studies would be necessary to elucidate the role of miR-135b in these processes during psoriasis development. Our results highlight the potential of this miRNA as a biomarker. Our data reveal that the basal levels of miRNA-135b in lesional skin could help to identify those patients that improve after treatment with biologics, regardless of the therapeutic target used.

## Figures and Tables

**Figure 1 cells-09-01603-f001:**
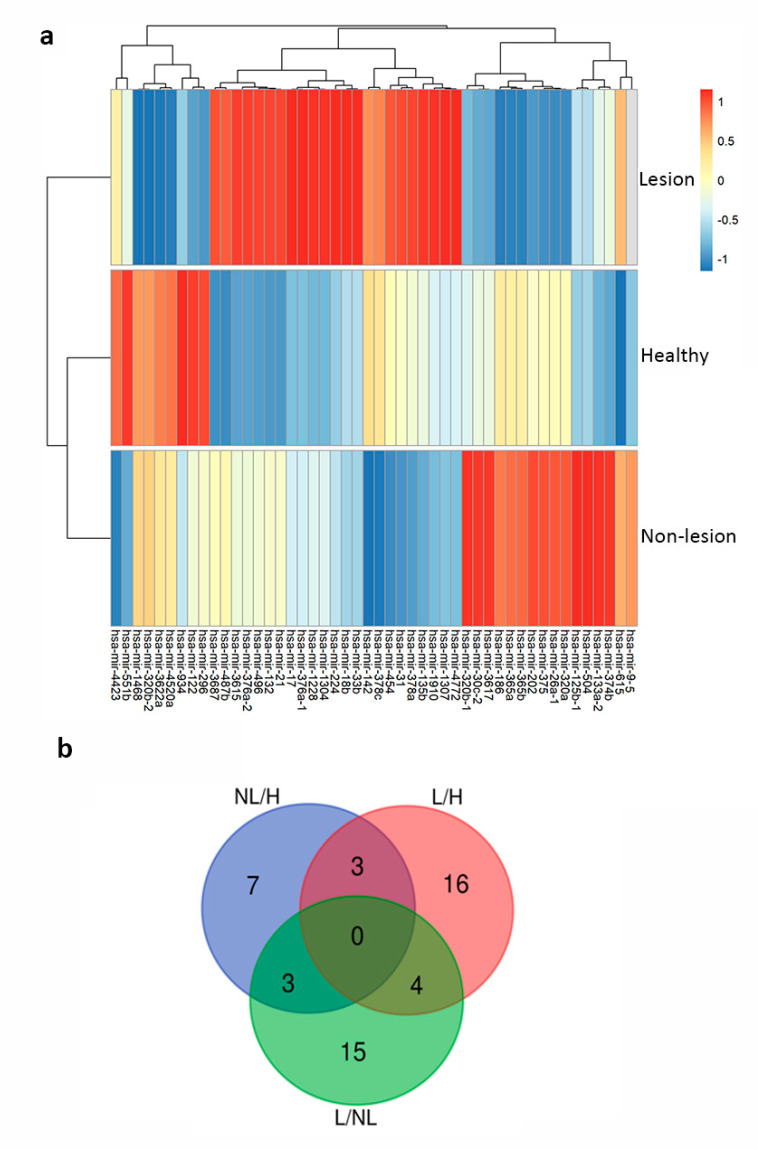
Differential expression of known miRNAs in healthy (H), non-lesional (NL) and lesional (L) psoriatic skin. (**a**) Heat map of hierarchical clustering of skin samples based on the 49 differentially expressed miRNAs. Each line represents z-score mean in the indicated group. (**b**) Venn diagram indicating the number of miRNAs that were > 2-fold differentially expressed in any of the three comparisons. Data correspond to skin samples from 5 psoriasis patients (lesional and non-lesional skin) and from 5 healthy controls.

**Figure 2 cells-09-01603-f002:**
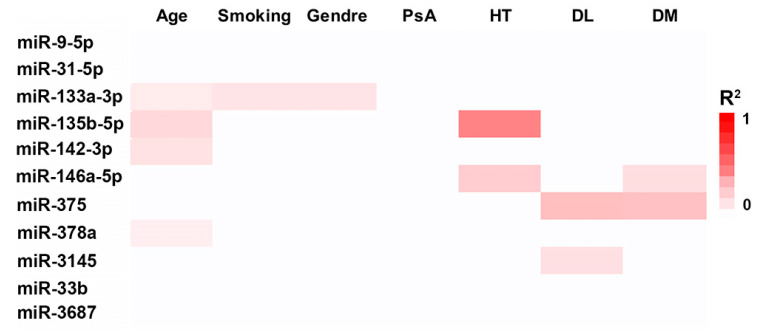
Effect of demographic and clinical parameters on the expression of miRNAs. Data were analyzed by a general linear model (GLM), only those significant values (*p* < 0.05) are shown. Psoriatic arthritis (PsA), arterial hypertension (HT), dyslipidemia (DL), diabetes mellitus (DM).

**Figure 3 cells-09-01603-f003:**
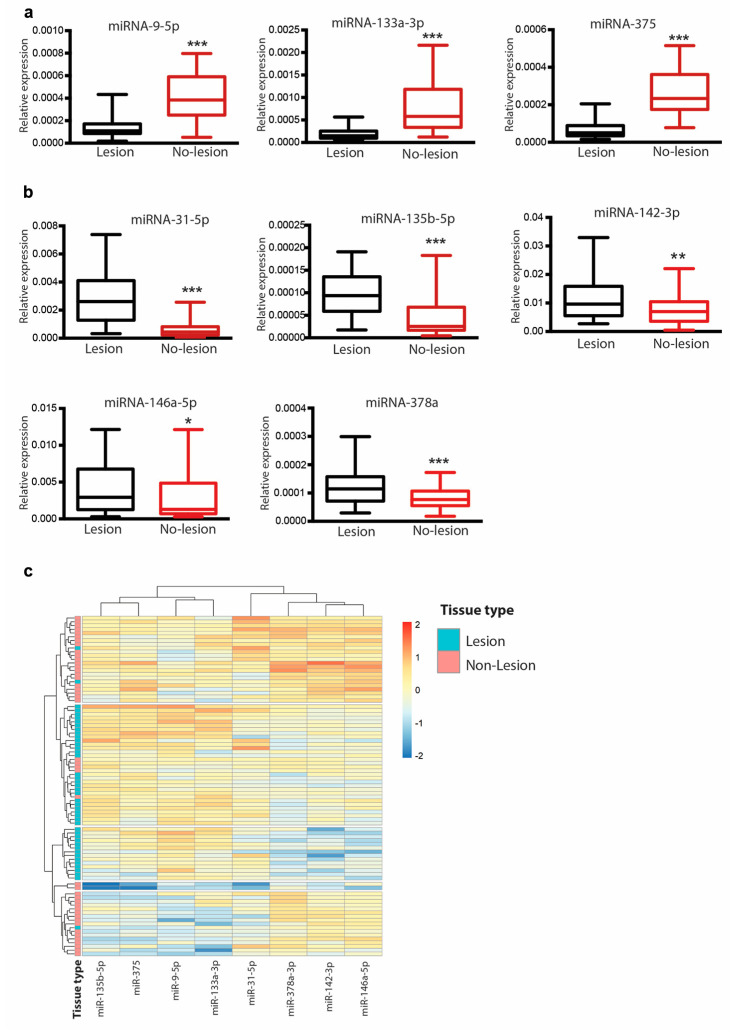
Quantitative PCR analysis of miRNAs in lesional and non-lesional psoriatic skin. (**a**) Differential expression of miRNA-9-5p, miRNA-133a-3p and miRNA-375 in lesional and non-lesional skin from 44 psoriatic patients. (**b**) Differential expression of miR-31-5p, miRNA-135b-5p, miRNA-142,3p, miRNA-146a-5p and miRNA-378a in lesional and non-lesional skin from 44 psoriatic patients. Data correspond to the relative levels of indicated miRNA with respect to the geomean of 5S and RNU1A1. Data were analyzed using paired t-test. * *p* <0.05, ** *p* < 0.001 *** *p* < 0.000. (**c**) Heat map of hierarchical clustering of skin samples based on the expression of validated miRNAs showed in (a).

**Figure 4 cells-09-01603-f004:**
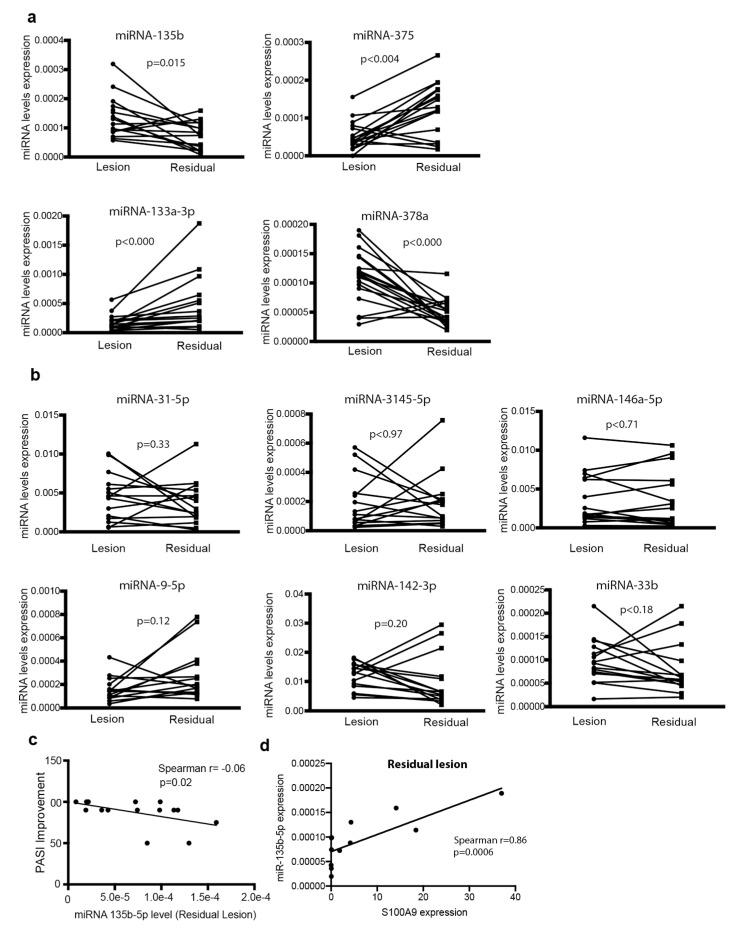
miRNA levels in lesional skin before treatment onset and in residual lesions after 3 months of treatment. (**a**) Expression of miRNA-135b-5p, miRNA-133a-3p, miRNA-375 and miRNA-78a in lesional skin before treatment onset and in residual lesions after 3 months of biological therapy (*n* = 15). (**b**) Expression of miR-31-5p, miR-3145-5p, miRNA-146a-5p, miR-9-5p, miR-142-3p and miR-33b as in (a). Lines indicate matched lesional and residual lesions from the same patient. Data correspond to the relative expression with respect to the geomean of 5S and RNU1A1 expression. Data were analyzed using Wilcoxon signed rank test. (**c**) Correlation between the levels of miRNA-135b in residual lesions and the disease improvement after 3 months of treatment. (**d**) Correlation between the levels of miR-135b and S100A9 in residual lesions. Data were analyzed using Spearman test.

**Figure 5 cells-09-01603-f005:**
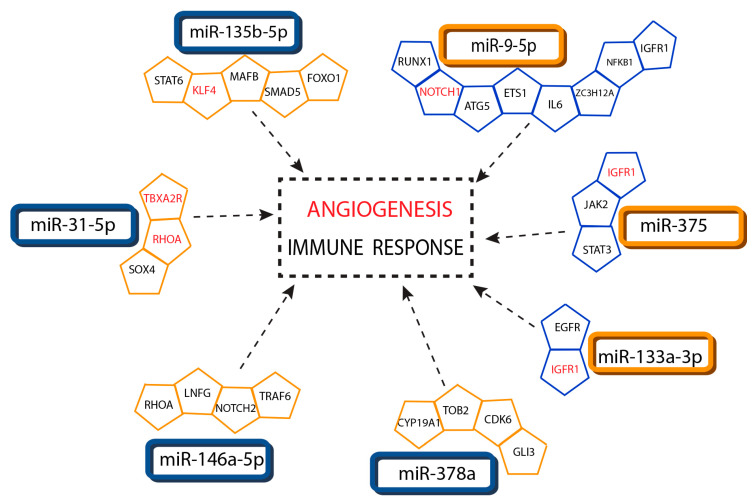
Interaction network of miRNAs and target mRNAs associated with immune response and angiogenesis. Networks showing interactions between differentially expressed miRNAs and differentially expressed messenger RNA targets associated with immune response and angiogenesis (identified in red) processes. Interactions were identified using miRTarBase and subjected to the PANTHER system analysis. Interacting pairs were filtered to keep only those with anti-correlated expression. Only targets with functional support are shown. Upregulated molecules are shown in blue and downregulated ones in orange.

**Table 1 cells-09-01603-t001:** Multiple regression analysis for Psoriasis Area and Severity Index (PASI).

	β Coefficient (95% CI)	*p*-Value
DM	−10.36 (−20.13–−0.59)	0.039
* creatinine levels	21.49 (8.21–34.7)	0.003
** miRNA-9-5p	7.35 (2.44–12.25)	0.005

DM: diabetes mellitus, *: basal levels, **: expression in lesional skin.

**Table 2 cells-09-01603-t002:** Multivariable logistic regression analyzing improvement of PASI after 3 months of treatment. All covariates with *p* value < 0.1 in the univariable analysis (Appendix A) were considered.

	OR (95% CI)	*p*-Value
**Treatment** * anti-IL12/IL23 anti-TNFa		
	0.31 (0.05–1.88)	0.200
	0.55 (0.08–3.83)	0.550
**Age**	0.95 (0.90–1.0)	0.078
**NL miRNA-146a**	2.33 (1.25–4.58)	0.015
**L miRNA-135b**	6.06 (1.57–23.33)	0.009

* Reference value anti-IL-17 treatment. Abbreviations: CI: confidence interval, NL: non-lesional, L: lesional, OR: odds ratio.

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
