# Peer review of "Expression of miR-135b in Psoriatic Skin and Its Association with Disease Improvement"

_cells, 2020, doi:10.3390/cells9071603_

Round 1

Reviewer 1 Report

The authors of the manuscript aimed to characterize the expression of miRNAs in psoriatic skin not previously described in psoriasis, the changes induced following the treatment with biologicals and their association with disease improvement. The authors performed small RNA NGS with SOLID assay (5 patients vs 5 controls, lesional vs nonlesional skin) and validate selected 12 miRNAs with 44 additional psoriasis patients using miRNA qPCR with Exiqon primers. In addition, samples after 3 months biological treatment were collected and compared in RT-qPCR analysis.

General comments: The manuscript is timely and addresses the important question about the association of miRNAs with biological treatment responses in the skin. Only 5 samples per each group were subjected to NGS, not all the methods are described sufficiently detailed, discussion as well as data analysis should be significantly improved.

Specific comments:

  • The abstract should contain information that NGS was also performed.
  • As NGS was performed only on 5 samples in each group, the dataset maybe biased. The authors should better discuss the limitations of the study.
  • lines 200-202: the authors report that 7 patients achieved PASI 100, 18 patients PASI 90, 4 patients PASI 75, 3 patients PASI 50 and only 1 patient improved a 25% from baseline. Please include % after each number as otherwise this sentence is confusing.
  • Figure 2C should be improved. Currently it is not possible to read all the symbols and numbers, at least in reviewer version PDF.
  • As RT-qPCR was done using geometric mean expression of 5S ribosomal RNA and RNU1A1 as reference, it would be informative to explain why this combination and how the two targets varied independently across the samples.
  • Figure 3C – It is not clear how authors can conclude from this result that miR-135b-5p is associated with disease improvement. As the relative levels of miR-135b-5p are in positive association with the disease severity score PASI, it is also possible that miR-135b-5p is simply associated with disease severity. The authors should also perform regression analysis between PASI and miR-135-5b and other selected miRNAs form the samples collected before the biological treatment and present the data.
  • The methods are not provided in sufficiently detailed. For example, the authors do not provide information how Figure 2 and 3 “relative expression” levels were calculated. Is the level each miRNA is comparable with other in these figures? Please improve.
  • General linear model demonstrates that miR-133a is higher in smokers and males. Could not it be purely that smokers are more prevalent among men?
  • To analyze the biological pathways that the group of differentially expressed miRNAs regulate, they application of IPA enables assess only miRNA-mRNA pairs with anti-correlated expression. As many miRNAs are regulated by the same pathways as targets and often both apparently upregulated, this approach is limiting. For example, this is probably the reason why IPA analysis reveals that miR-146a-5p regulates only GPM6B, and miR-135b-5p that changes with biological treatment regulates only CYP3A5. Additional analysis using other approaches, for example including all altered mRNAs should be used or limitations of the IPA analysis should be discussed.
  • The discussion that “the levels of miR-31, miR-142 and miR-146a in psoriatic skin do not change after treatment with biologics, in spite of the improvement of patients, suggests that their local expression is not related to the presence of inflammation, but with other processes preceding the inflammatory processes” does not seem to be relevant as the authors assess the situation after the treatment not before the inflammation takes place.

Author Response

Replies to Reviewer #1

The authors of the manuscript aimed to characterize the expression of miRNAs in psoriatic skin not previously described in psoriasis, the changes induced following the treatment with biologicals and their association with disease improvement. The authors performed small RNA bulmissNGS with SOLID assay (5 patients vs 5 controls, lesional vs nonlesional skin) and validate selected 12 miRNAs with 44 additional psoriasis patients using miRNA qPCR with Exiqon primers. In addition, samples after 3 months biological treatment were collected and compared in RT-qPCR analysis.

General comments: The manuscript is timely and addresses the important question about the association of miRNAs with biological treatment responses in the skin. Only 5 samples per each group were subjected to NGS, not all the methods are described sufficiently detailed, discussion as well as data analysis should be significantly improved.

Specific comments:

The abstract should contain information that NGS was also performed.

Following the reviewer’s suggestion, we have included the information about NGS assay at Abstract in the new revised version.

As NGS was performed only on 5 samples in each group, the dataset maybe biased. The authors should better discuss the limitations of the study.

We agree with the reviewer's comment that the number of samples in the NGS assay is small and therefore limits the identification of differentially expressed microRNAs between groups and the robustness of the differences detected. However, even with this small number of samples we identified a group of undescribed microRNAs in psoriasis that was the focus of our work. In the revised version of the manuscript we have included under Discussion section the limitation of the NGS findings due to the small number of samples analyzed.

lines 200-202: the authors report that 7 patients achieved PASI 100, 18 patients PASI 90, 4 patients PASI 75, 3 patients PASI 50 and only 1 patient improved a 25% from baseline. Please include % after each number as otherwise this sentence is confusing.

We appreciate the observation of the reviewer; accordingly in the revised version we have added the corresponding percent of patients to each PASI value.

Figure 2C should be improved. Currently it is not possible to read all the symbols and numbers, at least in reviewer version PDF.

We apologize for the poor quality of Figure 2C. This figure has been replaced in the revised version by another one with better resolution and higher size (New Figure 3C).

As RT-qPCR was done using geometric mean expression of 5S ribosomal RNA and RNU1A1 as reference, it would be informative to explain why this combination and how the two targets varied independently across the samples.

Reference genes were selected according to Genorm algorithm implemented in Biogazelle q-base software. Genorm is an algorithm extensively used to determine the most stable reference genes from a set of tested candidates. As putative candidate reference genes we initially selected U6, 5S and RNU1A1 among the genes recommended by Qiagen. Following Genorm results we identified that the combination of 5S and RNU1A1 had the best stability score across e samples as shown in the next figure.

Figure 3C – It is not clear how authors can conclude from this result that miR-135b-5p is associated with disease improvement. As the relative levels of miR-135b-5p are in positive association with the disease severity score PASI, it is also possible that miR-135b-5p is simply associated with disease severity. The authors should also perform regression analysis between PASI and miR-135-5b and other selected miRNAs form the samples collected before the biological treatment and present the data.

We thank the reviewer comment, we agree that the correlation graph between PASI and miR-135b levels before the treatment could be confusing and does not reflects the message we wanted to transmit. We used that graph to show that after three months of treatment the reduction in miR-135b levels occurs in those patients with better index of improvement. In the revised version of the manuscript we have replaced that graph for the correlation between miR-135b levels and the percent of improvement reached after 3 months of treatment (New Figure 4C).  In addition, we analyzed the association of miRNAs levels with the expression of S100A9 and IL-12 as indicator of local inflammation, miR-135b was among the miRNAs analyzed the only one associated with the expression of S100A9 (new figure 4d in the revised manuscript). These data support the notion that the decrease in miR-135b is associated with an improvement in the inflammatory process. Finally, we would like to remark that regression analysis between miRNAs levels before the treatment and PASI score showed that among the 12 miRNAs analysed, only the expression of miRNA-9-5p in lesional skin was associated with PASI values, these data are shown in Table 1.

The methods are not provided in sufficiently detailed. For example, the authors do not provide information how Figure 2 and 3 “relative expression” levels were calculated. Is the level each miRNA is comparable with other in these figures? Please improve.

Following reviewer suggestion this issue is now clarified under Material and Methods in the revised manuscript. In all cases the expression of miRNAS was calculated as the relative expression respect to the geometric mean of 5S and RNUA1 expression, thus the levels of each miRNAs are comparable among them.

General linear model demonstrates that miR-133a is higher in smokers and males. Could not it be purely that smokers are more prevalent among men?

Thank you for your comment. To ruled out this possibility, we performed a “chi square test”. Our statistical analysis shows that the smoking condition among female and male is not statistically significant as shown in the next chi square table.

To analyze the biological pathways that the group of differentially expressed miRNAs regulate, they application of IPA enables assess only miRNA-mRNA pairs with anti-correlated expression. As many miRNAs are regulated by the same pathways as targets and often both apparently upregulated, this approach is limiting. For example, this is probably the reason why IPA analysis reveals that miR-146a-5p regulates only GPM6B, and miR-135b-5p that changes with biological treatment regulates only CYP3A5. Additional analysis using other approaches, for example including all altered mRNAs should be used or limitations of the IPA analysis should be discussed.

Following the reviewer’s recommendation, we used an additional approach to identify biological processes associated with miRNA-targets. Using this approach we identified an enrichment in immune response and angiogenesis processes. miRNAs targets were identified using miRTarBase software selecting only those classified as “Functional”, then list of genes was subjected to PANTHER Classification System (http://pantherdb.org). As reference list, DE expressed genes in lesional psoriatic skin vs non-lesional skin from GEO (GSE67785) was used. These data are now shown in the New Figure 5 of the revised manuscript

The discussion that “the levels of miR-31, miR-142 and miR-146a in psoriatic skin do not change after treatment with biologics, in spite of the improvement of patients, suggests that their local expression is not related to the presence of inflammation, but with other processes preceding the inflammatory processes” does not seem to be relevant as the authors assess the situation after the treatment not before the inflammation takes place.

We agree that based in our results is ventured to suggest that miR-31, miR-142 and miR-146a are not related to the inflammation. The idea we wanted to expose is that if the increase in the expression of these miRNAs in the lesional skin was due to the inflammatory environment, their levels should have decreased after 3 months of treatment when the inflammation in most cases had decreased. We agree with the reviewer that it is very important to clarify this point, so we have included in the revised version of the manuscript the expression of IL-12 and S100A9 in skin samples as indicators of local inflammation (New supplementary Fig. 1a).

To clarify this point, we modified the sentence to “the levels of miR-31, miR-142 and miR-146a in psoriatic skin do not change after treatment with biologics in spite of the inflammation decrease, suggest that the high expression of these miRNAs in lesional skin is not associated with the inflammatory process”.

Reviewer 2 Report

In the article entitled “Expression of miR-135b in psoriatic skin and its association with disease improvement”, authors have performed an unbiased NGS analysis of skin samples from healthy donors and psoriatic patients. From this latter group of individuals, both lesioned and not lesioned areas of the skin were collected and compared in terms of miRNA expression level. In a validation phase, the expression of 12 selected miRNAs was analysed in new skin samples from 44 patients

with plaque psoriasis and in 15 patients an additional sample was obtained after 3 months of biological treatment. Authors identified a down-regulation of miR-9-5p, miR-133a-3p and miR-375 in the lesioned skin of psoriasis patients. After treatment, expression of miR-133a-3p, miR-375, miR-378a and miR-135b in residual lesions returned towards the levels observed in non-lesional skin. Furthermore, modulation of miR-135b was associated with patient improvement upon drug treatment and authors hypothesize a role for this miRNA in the pathogenesis of this disease. Abstract ends with the sentence: “The combined evaluation of miR-146a and miR-135b basal expression could be useful to predict the response to treatment.”

Although the paper describes interesting observations, the result section in particular needs to be substantially improved in order to make clear to the reader the pattern of experiments and results and the reasoning behind the planning of further steps of experimentation.

  1. NGS phase is very poorly described. It is actually absent from the main methods. Since it is at the base of miRNA selection for the validation phase, this flaw needs to be addressed. My opinion is that NGS results should be among the main figures. Venn diagram of supplemental Figure 1b shows how the different comparisons between the groups (healthy, lesioned and not lesioned skin) are overlapping in terms of differentially expressed (DE) miRNAs. How miRNAs should be instead clearly defined for their behavior among the three groups should be better defined i) in the text of the results; ii) in the heatmap for Figure S1. As an example, do miRNAs that increase in lesioned skin compared to not lesioned skin change between healthy and not lesioned skin? And how? In other words, does the not lesioned skin of psoriatic subjects show an intermediate expression of miRNAs dysregulated in psoriasis between the healthy and the lesioned skin? This must be described in more details. In Table S1, we can see the FC and the statistical significance for these differences, but it is not easy for the reader to come up with a pattern of miRNA modulation in the three groups. In particular, a color code for the three groups in the heatmap of Figure S1 may help decipher this specific point (to further help data representation, authors may z-score the expression level of single miRNAs, by normalizing heatmap rows). By the way, image quality of the heatmap is very poor, and I was not able to clearly read the column nomenclature.
  2. Authors should more clearly define how the selection for the validation phase was done. It is unintelligible.
  3. Authors normalize their RT-qPCR data Data using the geometric mean expression of 5S ribosomal RNA and RNU1A1. Is this a strategy they come up because it works fine for skin tissue? Any other reason? It should be better defined since normalization procedure is key to identify DE miRNAs.
  4. Authors show the effect of demographic and clinical variables on miRNA expression. What about multivariate analysis in which the DE miRNAs are “corrected” by relevant parameters? Possibly sample number is not high enough to allow the correction for all the multiple parameters they took in account, but actually sample number is sufficient to at least correct for age and sex.
  5. I wonder how much miRNA expression was different in different plaques from the same individual? Have authors ever controlled for this variability? On the same page, how miRNA differential expression is linked to differential cell distribution in the skin? Have authors tried to analyze the correspondent histology from the skin samples they evaluated by molecular biology? Are they aware of the possibility that some of their skin samples may contain high proportions of leukocytes? Is this described in the literature? They should at least discuss this point. In the discussion, FOXO1 is described to have a key role in the function and development of Treg cells. Are Treg cells present in the lesions? Are these cells usually increased upon drug treatment? This is a very important point, that needs to be better described.
  6. As above for the heatmap of Figure S1, authors should add a color code for lesional and non lesional in Figure 2C. Without a color code, the capability of DE miRNA to stratify different samples is not clear. I probably also missed the point of why only some of the DE miRNAs were used for supervised clustering of the samples. Were they the best in classification? This should be clearly explained.
  7. Association of interacting miRNA-targets with Psoriasis is very poorly described. Unless authors try and better explain why those target mRNAs are interesting on a biological point of view, the results seem a mere bioinformatics exercise.
  8. Authors cite both a potential pathogenetic role of the identified miRNAs in psoriasis and their potential use as prognostic biomarkers of treatment response. These are both very relevant issues and the presence of both weakens the message, in my opinion. What is the ultimate message authors want to leave the reader with? This should be better highlighted, not only in the discussion, but also in the abstract itself.
  9. Regarding biomarker development, I am wondering whether authors have quantified miRNAs in paired samples of serum/plasma. Do they envision a benefit in skin miRNA quantification? They should discuss this. Regarding the specific point, and the result they got on miR-146 not changing after treatment, authors may cite the paper “Extracellular MicroRNA Signature of Human Helper T Cell Subsets in Health and Autoimmunity” (Torri A, et al. J Biol Chem. 2017 doi: 10.1074/jbc.M116.769893) in which, consistently, miR-146a was not found decreased in serum of psoriatic patients after etanercept treatment.
  10. Abstract should better describe the study, since it should stand by itself. As an example, “Expression of 12 microRNAs was analysed in skin samples from 44 patients….”. Where these 12 microRNAs come from? It should be briefly described. Another example: “The combined evaluation of miR-146a and miR-135b basal expression could be useful to predict the response to treatment.” While reading the abstract one may ask himself where is miR-146a coming from? It was not named before this last sentence.

Author Response

Replies to Reviewer #2

In the article entitled “Expression of miR-135b in psoriatic skin and its association with disease improvement”, authors have performed an unbiased NGS analysis of skin samples from healthy donors and psoriatic patients. From this latter group of individuals, both lesioned and not lesioned areas of the skin were collected and compared in terms of miRNA expression level. In a validation phase, the expression of 12 selected miRNAs was analysed in new skin samples from 44 patients

with plaque psoriasis and in 15 patients an additional sample was obtained after 3 months of biological treatment. Authors identified a down-regulation of miR-9-5p, miR-133a-3p and miR-375 in the lesioned skin of psoriasis patients. After treatment, expression of miR-133a-3p, miR-375, miR-378a and miR-135b in residual lesions returned towards the levels observed in non-lesional skin. Furthermore, modulation of miR-135b was associated with patient improvement upon drug treatment and authors hypothesize a role for this miRNA in the pathogenesis of this disease. Abstract ends with the sentence: “The combined evaluation of miR-146a and miR-135b basal expression could be useful to predict the response to treatment.”

Although the paper describes interesting observations, the result section in particular needs to be substantially improved in order to make clear to the reader the pattern of experiments and results and the reasoning behind the planning of further steps of experimentation.

  1. NGS phase is very poorly described. It is actually absent from the main methods. Since it is at the base of miRNA selection for the validation phase, this flaw needs to be addressed. My opinion is that NGS results should be among the main figures. Venn diagram of supplemental Figure 1b shows how the different comparisons between the groups (healthy, lesioned and not lesioned skin) are overlapping in terms of differentially expressed (DE) miRNAs. How miRNAs should be instead clearly defined for their behavior among the three groups should be better defined i) in the text of the results; ii) in the heatmap for Figure S1. As an example, do miRNAs that increase in lesioned skin compared to not lesioned skin change between healthy and not lesioned skin? And how? In other words, does the not lesioned skin of psoriatic subjects show an intermediate expression of miRNAs dysregulated in psoriasis between the healthy and the lesioned skin? This must be described in more details. In Table S1, we can see the FC and the statistical significance for these differences, but it is not easy for the reader to come up with a pattern of miRNA modulation in the three groups. In particular, a color code for the three groups in the heatmap of Figure S1 may help decipher this specific point (to further help data representation, authors may z-score the expression level of single miRNAs, by normalizing heatmap rows). By the way, image quality of the heatmap is very poor, and I was not able to clearly read the column nomenclature.

Following the reviewer’s suggestion NGS data are now shown as main figure (New Figure 1 in the revised manuscript), and they have been included in the main Methods. ­­­­­ We agree with the reviewer that as we showed the data it was not easy to observe the differences between the samples of lesion, non-lesion and healthy subjects. In order to make it easier for the reader to see these differences we have replaced the scheme with a heatmap using the average z score of each miRNA. In this new heatmap the dendogram indicates a greater similarity between the samples of healthy subjects and non-lesional skin of psoriais, while the samples of lesions appear more distant.  We expected that samples from non-lesional psoriatic skin had showed an intermediate expression of miRNAs between the healthy and the lesioned skin. We attribute this result to the small number of samples included for NGS. These data are discussed in the revised version of the manuscript. We apologize for the poor quality of the previous heatmap, in the revised version we have improved the resolution of corresponding image.

2. Authors should more clearly define how the selection for the validation phase was done. It is unintelligible.

We apologize for the lack of a clear explanation of this important issue. In this regard, the selection of miRNAs for the validation phase is explained in more detail in the revised version. From the NGS data we selected 8 MicroRNAs representative of each comparison group (L vs NL, L vs H, or NL vs H) that had not been previously validated in psoriasis. We decided include also 4 miRNAs previously described in psoriasis in the validation phase as internal controls.

3. Authors normalize their RT-qPCR data Data using the geometric mean expression of 5S ribosomal RNA and RNU1A1. Is this a strategy they come up because it works fine for skin tissue? Any other reason? It should be better defined since normalization procedure is key to identify DE miRNAs.

We agree with the reviewer about the importance of normalization in the studies of microRNAs. Reference genes were selected according to Genorm algorithm implemented in Biogazelle q-base software. Genorm is an algorithm extensively used to determine the most stable reference genes from a set of tested candidates. As putative candidate reference genes we initially selected U6, 5S and RNU1A1 among the genes recommended by Qiagen. Following Genorm results we identified that the combination of 5S and RNU1A1 had the best stability score across e samples as shown in the next figure.

4. Authors show the effect of demographic and clinical variables on miRNA expression. What about multivariate analysis in which the DE miRNAs are “corrected” by relevant parameters? Possibly sample number is not high enough to allow the correction for all the multiple parameters they took in account, but actually sample number is sufficient to at least correct for age and sex.

The reviewer brings up a relevant point here. Indeed, we corrected our multivariate analyses by the potential confounding factors. To do so, we performed univariate analyses in which we analysed the association between miRNA expression and each of the variables considered in this study (including sex and age), and only variables with a significant association in these univariate tests were considered in multivariate analyses. We are also aware than some variables could have an effect in some of the predictors used in multivariate analyses rather than in the response variable. To consider for this possibility, our multivariate models accounted for potential collinearity between variables. At any rate, miRNA levels were corrected for not more than 3 confounding factors.

5. I wonder how much miRNA expression was different in different plaques from the same individual? Have authors ever controlled for this variability? On the same page, how miRNA differential expression is linked to differential cell distribution in the skin? Have authors tried to analyze the correspondent histology from the skin samples they evaluated by molecular biology? Are they aware of the possibility that some of their skin samples may contain high proportions of leukocytes? Is this described in the literature? They should at least discuss this point. In the discussion, FOXO1 is described to have a key role in the function and development of Treg cells. Are Treg cells present in the lesions? Are these cells usually increased upon drug treatment? This is a very important point, that needs to be better described.

We thank the reviewer for these valuable comments. Regarding the first question, we apologize for the misunderstanding, differences between Lesion and non-lesion samples were always analyzed as paired samples. However, we decided to represent the data as boxplot to show the degree of data dispersion.  The reviewer’s comment regarding the differences in leukocytes is very interesting. It is known that lesional skin is characterized by an important inflammatory infiltrate and hyperkeratosis among other histological characteristics. Although the leukocyte infiltration was not evaluated in our samples, we analyzed the expression of IL-12 and S100A9 as an approach to detect differences in inflammation levels between the different samples. These data are now included in the revised version.  As expected, both genes were upregulated in lesional skin compared to non-lesional skin and residual lesions. However, no correlation between the expression of these molecules and disease severity was observed. We also analyze the association of miRNAs expression with the levels IL-12 and S100A9.  Expression of miR-135b correlated with S100A9 in lesional skin and also in residual lesions, but in non-lesional skin. These data support the role of miR-135b in the inflammatory response of psoriasis. These data are shown in the new Figure 4d and new Supplementary Figure 1 in the revised manuscript.

Dysfunctional or reduced Treg cells has been described in peripheral blood and in psoriatic lesional skin in patients (Sugiyama 2005, Keijser 2013, Kotb I.S. BJD 2018 DOI 10.1111/bjd.16336). Using the murine imiquimod-induced model of psoriasis, it was demonstrated that Treg cells limit the exacerbation of local skin inflammation and initiate also disease remission (Cell Rep 2018 25:3564-3572). Regarding the modification of Treg cells after psoriasis treatment, it has been described that the frequency of Treg cells increases in the skin of psoriasis patients after topical treatment with steroids (J Derm Sci 2008 51:200-203). In addition, administration of anti- IL-17A or IL-23p19 antiboides in a psoriasis murine model induced a significant increase in the number or Foxp3+ IL-10+ Treg cells (J Dermatol Sci 2019;95-90-98). In human being the treatment with anti-TNF-a drugs induced the up-regulation of circulating Treg, this increase in Treg cell correlates with reduction in the disease severity score (Eur J Dermatol 2011; 21:344-8). These explanation have been included under Discussion in the revised version.

6. As above for the heatmap of Figure S1, authors should add a color code for lesional and non lesional in Figure 2C. Without a color code, the capability of DE miRNA to stratify different samples is not clear. I probably also missed the point of why only some of the DE miRNAs were used for supervised clustering of the samples. Were they the best in classification? This should be clearly explained.

Heatmap in Figure 2c has been replaced in the New figure 3c, we added a color code  to distinguish lesion vs non-lesion samples. In this new heatmap, all validated miRNAs are represented

7. Association of interacting miRNA-targets with Psoriasis is very poorly described. Unless authors try and better explain why those target mRNAs are interesting on a biological point of view, the results seem a mere bioinformatics exercise.

We appreciate this comment and agree that this was a weakness in our data. Because the literature review did not show any direct links of most of the target identified by IPA software with the pathogenesis of psoriasis, we used an additional approach to identify the association of miRNA-targets with this disease. Functional miRNA-targets were identified using the miRTarBase software, genes were filtered to keep only those with anti-correlated expression and subjected to PANTHER Classification System to determine the over-representation of biological processes annotated in Gene Ontology. 

We identified an enrichment of targets in immune response and angiogenesis processes that are sown now in new Figure 5, and commented in more detail under Discussion in the revised manuscript

8. Authors cite both a potential pathogenetic role of the identified miRNAs in psoriasis and their potential use as prognostic biomarkers of treatment response. These are both very relevant issues and the presence of both weakens the message, in my opinion. What is the ultimate message authors want to leave the reader with? This should be better highlighted, not only in the discussion, but also in the abstract itself.

Thank you for your comment, we think that the main message of our work is the potential use as biomarkers; and this message has been conveyed in the revised manuscript.

9. Regarding biomarker development, I am wondering whether authors have quantified miRNAs in paired samples of serum/plasma. Do they envision a benefit in skin miRNA quantification? They should discuss this. Regarding the specific point, and the result they got on miR-146 not changing after treatment, authors may cite the paper “Extracellular MicroRNA Signature of Human Helper T Cell Subsets in Health and Autoimmunity” (Torri A, et al. J Biol Chem. 2017 doi: 10.1074/jbc.M116.769893) in which, consistently, miR-146a was not found decreased in serum of psoriatic patients after etanercept treatment.

This is a very interesting question; indeed we initially measured the expression of free miRNAs in serum samples, however, we observed a high variability in the miRNA levels across different cohorts of patients presumably because normalization strategy in the absence of known stable miRNAs. Following the reviewer’s suggestion, we have cited and discussed the work from Torri A et al.

10. Abstract should better describe the study, since it should stand by itself. As an example, “Expression of 12 microRNAs was analysed in skin samples from 44 patients….”. Where these 12 microRNAs come from? It should be briefly described. Another example: “The combined evaluation of miR-146a and miR-135b basal expression could be useful to predict the response to treatment.” While reading the abstract one may ask himself where is miR-146a coming from? It was not named before this last sentence.

We appreciate the reviewer comment. Following his/her suggestion, the abstract  has been modified in the revised version.

Reviewer 3 Report

The author described the miR-135b expression was associated with the improvement of psoriasis. The correlation was very clear and very promising, could be used as a biomarker.

But the underlying mechanism is not fully understood. The positive control miR-146a was a well-known miRNAs that play vital role in inflammation response. miR-135b might affect inflammation. It should be investigated how miR-135b is related with psoriasis.  At least the predicted target cyp3a5 they indicate should be studied, including target confirmation, and the association between cyp3a5 and psoriasis. 

Author Response

Replies to Reviewer 3.

The author described the miR-135b expression was associated with the improvement of psoriasis. The correlation was very clear and very promising, could be used as a biomarker.

But the underlying mechanism is not fully understood. The positive control miR-146a was a well-known miRNAs that play vital role in inflammation response. miR-135b might affect inflammation. It should be investigated how miR-135b is related with psoriasis.  At least the predicted target cyp3a5 they indicate should be studied, including target confirmation, and the association between cyp3a5 and psoriasis. 

We appreciate the positive comment from the reviewer. Because the 2 additional reviewers also requested to improve this issue too, we used an additional approach to identify the association of miRNA-targets with psoriasis. Functional miRNA-targets were identified using the miRTarBase software, genes were filtered to keep only those with anti-correlated expression and subjected to PANTHER Classification System to determine the over-representation of biological processes annotated in Gene Ontology.  We identified an enrichment of targets in immune response and angiogenesis processes that are sown now in new Figure 5. In this new approach we included only validated targets supported by functional assays.

Round 2

Reviewer 2 Report

The authors have significantly improved the manuscript and result presentation. Few things need to be further addressed. 

Heatmap of Figure 3C. The z score should have been applied to single miRNAs, not to single samples. In other words, the applied normalization now shows how some miRNAs are more expressed than others (not very informative), while it should show how a single miRNA is more or less expressed in different samples (informative for the study). 

Lines 307-311. The consistent presence of miR-146a upon therapy is not telling us that the miRNA is not involved in the inflammatory process. As already discussed before, it may also relate to the anti-inflammatory process (possibly Treg-mediated) that prevails upon biological treatment. Since indeed miR-146a is known to be very highly expressed by Treg and also fundamental for their function, this hypothesis is not weak. I suggest to rephrase to better discuss the significance of the result. 

The manuscript discussion ends with sentences around other than miR-135b miRNAs (miR-133 and miR-375). This is unexpected. It really sounds as the manuscript was interrupted. Please, try and conclude the manuscript with final considerations about the miRNA at the center of the study. 

Minor points:

please be aware of grammar. Examples: line 144 this miRNAs; line 195 these miRNA. 

please be aware that ref.21 is possibly not the intended one (the suggested Torri et al.).

Author Response

Response to Reviewer #2

The authors have significantly improved the manuscript and result presentation. Few things need to be further addressed. 

Heatmap of Figure 3C. The z score should have been applied to single miRNAs, not to single samples. In other words, the applied normalization now shows how some miRNAs are more expressed than others (not very informative), while it should show how a single miRNA is more or less expressed in different samples (informative for the study). 

Following the reviewer’s suggestion heatmap of Figure 3C has been replaced by a new one following the suggestion of the reviewer.

Lines 307-311. The consistent presence of miR-146a upon therapy is not telling us that the miRNA is not involved in the inflammatory process. As already discussed before, it may also relate to the anti-inflammatory process (possibly Treg-mediated) that prevails upon biological treatment. Since indeed miR-146a is known to be very highly expressed by Treg and also fundamental for their function, this hypothesis is not weak. I suggest to rephrase to better discuss the significance of the result. 

We appreciate the reviewer’s comment. We have rephrased this comment of the presence of miR-146a after the treatment, to better convey the significance of results.

The manuscript discussion ends with sentences around other than miR-135b miRNAs (miR-133 and miR-375). This is unexpected. It really sounds as the manuscript was interrupted. Please, try and conclude the manuscript with final considerations about the miRNA at the center of the study. 

We appreciate this comment, and the order of Discussion section has been modified to conclude the manuscript with the considerations about miR-135b

Minor points:

please be aware of grammar. Examples: line 144 this miRNAs; line 195 these miRNA. 

please be aware that ref.21 is possibly not the intended one (the suggested Torri et al.).

Thank you for your comments, grammar has been revised and references carefully checked and replaced when were incorrect.

Reviewer 3 Report

The functional study is still a weak part of the manuscript. The authors should at least demonstrate whether the mRNA levels of the predicated targets in Figure 5 are correlated with miRNA levels in lesional skin before treatment onset and in residual lesions. 

The data will be solid if they overexpress miR-135b and its mutant (seed sequence mutant) in cells and investigate whether the targets are altered.    

Author Response

Response to Reviewer 3.

The functional study is still a weak part of the manuscript. The authors should at least demonstrate whether the mRNA levels of the predicated targets in Figure 5 are correlated with miRNA levels in lesional skin before treatment onset and in residual lesions. 

The data will be solid if they overexpress miR-135b and its mutant (seed sequence mutant) in cells and investigate whether the targets are altered.   

We appreciate the reviewer’s comment and we have taken into account his/her concerns that are now well taken under Discussion section (page 12, lines 332-334 and 356-357). As we quote in the Round 1 of revision, we agree that this a weak part of our manuscript. However, all the targets we included in the last version have been validated by other authors with sufficient experimentation to classify them as Functional validated targets. Unfortunately, the additional assays requested by the reviewer are unfeasible to be addressed in a short period of time. We would need to get new skin samples to evaluate the expression of targets in parallel with miRNAs in order to perform the correlation analyses. We would like to remark that the functional study of miRNAs is out the scope of the present manuscript and the functional role of miR-135b in psoriasis development deserves a completely new independent piece of work.
